# Short Communication: Challenges and Applications of Structure-from-Motion Photogrammetry in a Physical Model of a Braided River

Pauline Leduc [1], Sarah Peirce [1], and Peter Ashmore [1]

[1]Department of Geography, The University of Western Ontario, London, N6A 3K7, Canada

**Correspondence:** Peter Ashmore (pashmore@uwo.ca)

**Abstract.** Extending the applications of Structure-from-Motion (SfM) photogrammetry in river flumes, we present the main challenges and methods used to collect a large dataset (> 1000 digital elevation models) of high-quality topographic data using close-range SfM photogrammetry with a resulting vertical precision of $\sim$ 1mm. Automatic target-detection, batch processing, and considerations for image quality were fundamental to successful implementation of SfM on such a large dataset, which was used primarily for capturing details of gravel-bed braided river morphodynamics and sedimentology. While the applications of close-range SfM photogrammetry are numerous, we include sample results from DEM differencing, which was used to quantify morphology change and provide estimates of water depth in braided rivers, as well as image analysis for mapping bed surface texture. These methods and results contribute to the growing field of SfM applications in geomorphology and close-range experimental settings in general.

## 1 Introduction

Photogrammetric techniques have a long history in geomorphology, both in the field and laboratory, but the emergence of "Structure-from-Motion" (SfM) digital photogrammetry represents a technological revolution in geomorphological terrain analysis (Westoby et al., 2012; Tarolli, 2014; Bakker and Lane, 2017; Javernick et al., 2014; Woodget et al., 2015). Unlike traditional methods which require a high level of expertise, a priori knowledge of camera positions, fixed and calibrated camera geometry, and/or the real-world 3D locations of ground control points (GCP), SfM allows camera positions and the geometry of a scene to be solved automatically and simultaneously (Westoby et al., 2012; Fonstad et al., 2013; Smith et al., 2016). In addition, the availability of inexpensive high-resolution digital cameras and user-friendly photogrammetric software to produce digital elevation models (DEMs) means that the resolution and quality of the DEMs is now primarily limited by quality of the input imagery (Chandler, 1999; Brasington and Smart, 2003; Rumsby et al., 2008). Fluvial geomorphologists are taking advantage of these advances and have used SfM photogrammetry to study rivers from large, dynamic braided rivers in the field to laboratory flumes and physical models (Kasprak et al., 2015; Leduc et al., 2015; Bakker and Lane, 2017; Morgan et al., 2016).While much of the research on SfM has been field-based (typically using unmanned aerial vehicle, UAV, platforms) recent reports show that SfM techniques have the potential provide a less expensive, but effective alternative to other methods such as laser scanning in close-range flume and laboratory settings (Kasprak et al., 2015; Morgan et al., 2016).

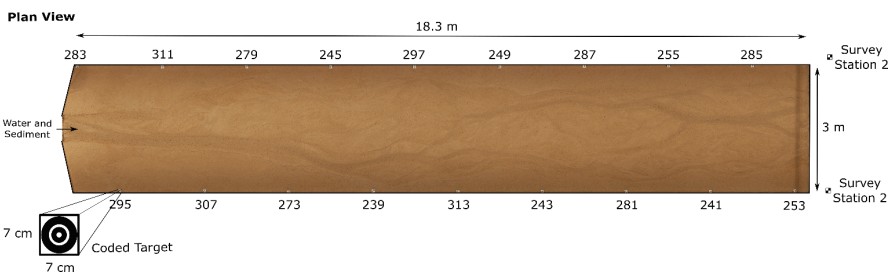

**Figure 1.** Planform view of the flume showing coded target locations and total station survey locations. Numbers refer to the unique target identifiers used in Agisoft PhotoScan's automated target detection.

Here, we present methods for DEM and orthophotos acquisition from a Froude-scaled physical model of a gravel-bed braided river. We used close-range SfM techniques, enhanced with custom scripts for automatic control target detection and batch processing, to collect over 1000 high-quality DEMs of the 18.3 x 3 m model surface over a series of braided river experiments. While general guidelines for using close-range SfM have been discussed elsewhere (see Morgan et al. (2016)), here we address specific challenges faced and present methods used to improve data collection and the resulting data quality. We demonstrate that these techniques can be used to extract detailed morphological information, water surface topography and flow depth, as well as grain size/texture data from braided river models. These efforts contribute to the identified need for ongoing learning about application and quality of SfM in laboratory settings (Morgan et al., 2016).

## 2  Physical Model and Experimental Procedure

Data was gathered from small Froude-scaled physical models of braided gravel-bed rivers in a river modelling flume located at the The University of Western Ontario (UWO) (Fig. 1). The flume was 18.3 m long and 3 m wide with adjustable slope and discharge with a maximum of 2.5 % and 2.7 $l.s^{-1}$, respectively. The grain size distribution ranged from 0.18 mm to 8 mm, with $D_{10}$ of 0.32 mm, $D_{50}$ of 1.18 mm, $D_{90}$ of 3.52 mm and a geometric standard deviation of 1.4 mm, representing the particle size distribution of the gravel fraction of a real gravel-bed braided river at an approximately 1:35 scale. The results presented come from a series of experiments covering six different stream power conditions to monitor morphological processes and variability over time. These experimental conditions extend the work of Morgan et al. (2016) into additional complex braided morphologies and graded grain size distributions.

Digital images of the model surface and bed topography were acquired from the drained bed (no standing water) at regular intervals of either 15 or 30 minutes, across six experiments that lasted between 29 and 68 hours each. Two sets of digital images (i.e. photo surveys, by imaging the full length of the flume and then repeating) of the drained model surface were taken for every interval using two Canon T5i cameras (18 mega-pixel sensor with 20 mm lenses) stationed on a movable trolley. The trolley was situated on horizontal rails 2.7 - 2.9 m (depending on flume slope and image location) above the model surface (Fig. 2) providing image coverage of the entire flume width. The cameras were positioned in a convergent geometry so that

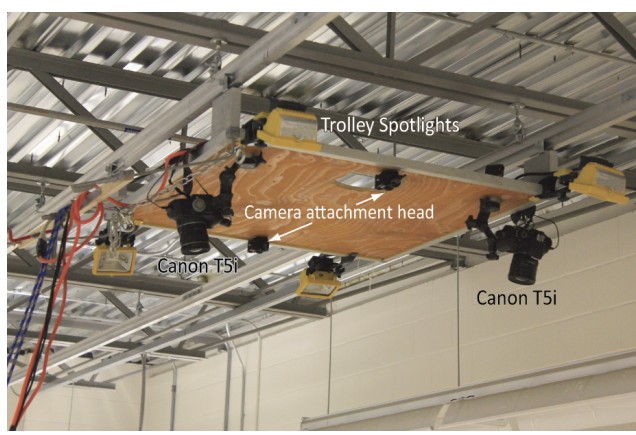

**Figure 2.** Movable trolley above model surface with 2 Canon cameras in a convergent position as well as four spotlights.

there was $\sim 80$ % transverse overlap between photos over the center area of the model where morphological change was expected to be greatest. The trolley was pulled along the length of the flume with a longitudinal (forward) image overlap of $\sim 60\%$ across an average of 100 photos (50 photos from each camera) to cover the flume area. The cameras were triggered simultaneously using the software DigiCamControl, which also allowed images to be downloaded directly to a computer. This

camera positioning and geometry was consistent throughout all experimental runs following a more traditional near-vertical aerial photography geometry (Gardner and Ashmore, 2011; Kasprak et al., 2015; Leduc et al., 2015) than is sometimes the case for SfM applications, which may use images from multiple positions and angles (Morgan et al., 2016). The two sets of digital images which don't have the same exact number of pictures, exact start and end locations and overlap, were used to estimate the precision of the survey (see section 3.1.1).

In addition to the dry bed photo surveys, additional wet bed photo surveys were acquired immediately prior to turning off the flow, when there was still water flowing in the model. These images were used to explore whether SfM could be used to map water surface topography in braided channels. During all photo surveys, spotlights attached to the camera trolley (Fig. 2) were used as the only light source to create consistent illumination of the model surface and minimize shadows and reflections that can negatively impact photogrammetric outcomes.

**3   DEM generation using SfM Digital Photogrammetry**

The software package Agisoft PhotoScan 1.0.0.1 (i.e. PhotoScan) was used for digital photogrammetric processing to convert both the dry and wet bed photo surveys into a high-resolution DEM and orthophotos. While SfM allows for the creation of a dense point cloud without a priori knowledge of camera or target locations, reference to "real-world" position still requires independent ground control points for georeferencing (Fonstad et al., 2013). Therefore, a dense control target array used 18

7 x 7 cm coded targets printed from Agisoft PhotoScan software placed on the inside walls of the flume via industrial Velcro (Fig. 1). Target locations were independently surveyed for each of the six experiments using a total station from two survey

| Experiment | $\mu$ (mm) | $\sigma$ (mm) |
|:---:|:---:|:---:|
| 1 | 0 | 0.4 |
| 4 | 0 | 0.2 |
| 9 | - 1 | 0.2 |
| 11 | - 1 | 0.6 |
| 12 | 0 | 0.3 |
| 13 | 0 | 0.2 |

**Table 1.** The mean value ($\mu$) and the standard deviation ($\sigma$) of the vertical difference between duplicate DEMs for each experiment

station locations at the downstream end of the flume (Fig. 1) and converted into a text file of 3D (xyz) positions (sub-millimetre precision from repeat surveys) using 3D intersection. The target coordinates were used in the automatic target detection process in PhotoScan. In future, it may be useful to have additional targets used as check points during processing. This process was used to generate DEMs from a dense cloud with a density of 80 points/$cm^2$ using the PhotoScan interpolation. The final cell size was 1.5 mm (close to the median grain size) and more than 1000 DEMs and orthophotos were generated.

### 3.1 Precision and error qualification from the DEM

Estimates of the vertical precision in the DEMs in each experiment were calculated from multiple photo surveys of the same surface, and non-moving, flat areas. Elevation accuracy was also assessed by direct comparison with a local laser scan of a small area of the model surface.

### 3.1.1 Multiple photo surveys of the same surface

Two sets of dry bed photo surveys (approximately 100 pictures for each set) were taken of the model surface at the end of each experimental run, and the resulting paired DEMs were used to estimate the mean and standard deviation of the vertical precision (Table 1). In addition to the precision quantification, the two DEMs of the same surface were compared so that only one DEM from each pair would be used for further analysis. For each pair of DEMs, a DEM of difference (DoD) was created. When the mean value of the DoD was less than 0.5 mm, preference was arbitrarily given to the first photo survey and DEM. If the difference in the paired DoD was greater than 0.5 mm, each DEM was then compared to the DEM from the previous time interval (for which the error decision was already made) and preference was given to the DEM providing the lowest mean value of elevation difference. Standard deviation was also used to detect significant errors (e.g. tilting and local artifacts) but the mean proved more useful in general because of the small standard deviation in elevation for most DEMs.

### 3.1.2 Non moving, flat areas

Flat, non-moving areas on the edges of the model surface, which were not reworked by the flow during the experiment, were also used to estimate the vertical precision across all DEMs within each experiment. This gives an estimate of precision for

| Experiment | $\sigma$ (mm) |
|:---:|:---:|
| 1 | 2.4 |
| 4 | 1.3 |
| 9 | 1.66 |
| 11 | 1.15 |
| 12 | 0.96 |
| 13 | 0.79 |

**Table 2.** Vertical precision estimates for each experiment based on the standard deviation ($\sigma$) in the distribution of the elevations for the non-moving areas.

surfaces that are known not to have changed elevation between surveys and gives information on repeatability for the surveys. The edge of the flat areas was defined by automatically detecting the slope break between the flat area and the channel bank in each row of the DEM. The error estimate was then calculated by differencing only the flat areas between two consecutive DEMs, and then merging the values within each experiment (Table 2). Based on this analysis, the overall DEM noise was around 1 mm, but noise was reduced in later experiments as data collection technique improved (Table 2).

### 3.1.3   Laser scan topography comparison

A final assessment of the data error compared a DEM produced from Agisoft PhotoScan to a DEM generated from a hand-held 3D surface laser scan (Exascan scanner from Creaform, with a resolution of 0.050 mm and the accuracy up to 0.040 mm.) of the same surface. The area scanned was about 30 cm * 39 cm which corresponds to over $50 * 10^3$ points. Figure 3 shows that the elevation distribution is roughly centred around 0 (mode= -0.08 mm) and fits a normal law ($\sigma = 0.62$, $\mu = -0.25$). Based on the distribution proprieties, the 99.7 % confidence interval of the difference is [-0.13 mm, 1.37 mm], which is again on the scale of the $D_{50}$ of the grain size in the flume.

## 4   Challenges: Improving data collection and outcomes in a laboratory flume

While SfM offers speed, accuracy, and flexibility, there were several challenges encountered when applying close-range SfM techniques in the laboratory. As a result, the quality of the DEMs within and between the experiments was inconsistent. Based on Table 2, in which the experiments are numbered in the order that they were completed, the data collection procedure and quality of the resulting DEMs improved with experience and better understanding of the influences on DEM quality.

### 4.1   The doming effect

Initial tests yielded some longitudinal doming in the DEMs which is often referred to in SfM literature (e.g., Kasprak et al., 2015; Smith et al., 2016; Morgan et al., 2016; James et al., 2017). A careful camera calibration using Agisoft Lens in addition to

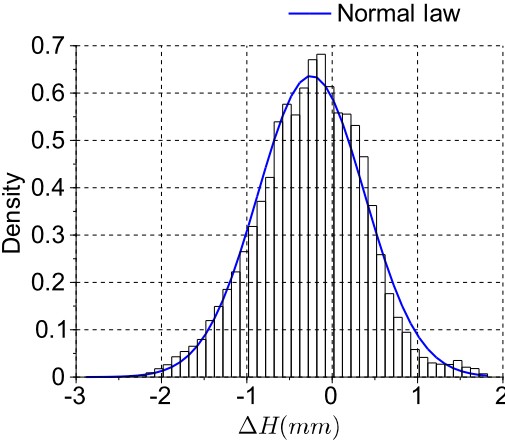

**Figure 3.** The elevation difference distribution comparing the DEM from Agisoft and the DEM generated using a Laser scan. The mean value is -0.25 mm (the mean absolute value is 0.53 mm) and the standard deviation is 0.62 mm (the mean absolute standard deviation is 0.43 mm).

a stronger image convergence using the two cameras (James and Robson, 2014) eliminated the doming effect and no systematic error was noticed afterwards.

## 4.2 Target detection and camera settings

While SfM does not require target detection for dense-point cloud generation, the overall data quality was affected by the
number of coded targets recognised by PhotoScan's automated target detection during data processing. It was important to maximize the number of targets detected by adjusting target and camera positions accordingly, and continually confirming that targets were being detected through visual analysis of each photo surveys. The convergent geometry of the two camera system may also be part of this solution, as has been reported in the past (Wackrow and Chandler, 2008; Smith et al., 2016) but we did not explicitly test this.

Image quality was also crucial to maximizing precision, as has been described (Mosbrucker et al., 2017). Superior camera focus was fundamental to SfM success and even a very slight softness in focus degraded DEM results considerably (Fig. 4) and in some cases made DEM results unusable. A fixed focal length lens is commonly recommended to maximise internal geometric stability (e.g., Mosbrucker et al., 2017). The cameras were also set to a fixed manual focus (rather than auto focus) but focus still slipped slightly at times and we realised that taping the focus ring was necessary. In the low light conditions
in the flume, camera aperture had to be large (images were taken at f3.5 and 1/8 second) and this may have reduced depth of field, considering that camera-object distance changed systematically along the flume (because of flume slope) by up to 30 cm. Even when focus was superficially good we discovered problems on the fairly-uniform sand surface and we found that unless close and careful attention was paid to this issue results could be downgraded considerably and large numbers of DEMs

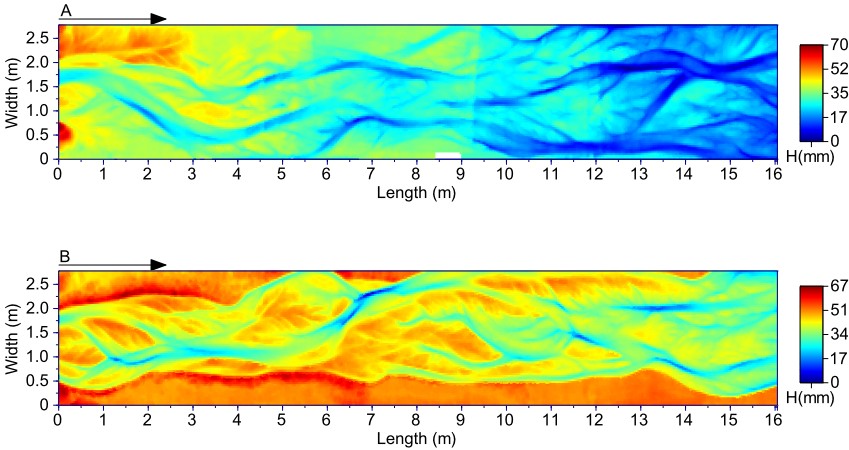

**Figure 4.** DEMs resulting from slightly out of focus images (A) compared to a DEM created under the same flow condition with superior focus (B).

could be lost (Fig. 4). In later experiments, the focus was improved during every photo survey by zooming in on small vector drawings placed in the field of view on the captured images, checking focus, and adjusting and re-imaging if necessary. Finally, capturing two photo surveys for each surface improved the probability of acquiring at least one set of high quality images and overall improved DEM results.

### 4.3   Processing time

Each photo survey (i.e., one set of images of the full flume length) took approximately 15 minutes to collect and approximately 5 hours of processing time in PhotoScan to generate a high-resolution DEM and orthophoto. To ensure that the data were processed continually, a simple Python script was written that allowed for batch processing of the photo surveys. The input for the script was the images from the photo surveys, coded target locations, and initial camera calibration parameters derived by PhotoScan. While the processing was time consuming, the automation made it possible to continuously process photos and generate >1000 high-resolution DEMs ($\sim$ 500 unique model surfaces) across all six experiments over a few months which was at least an order of magnitude faster than manual target acquisition and processing with older digital photogrammetry software (e.g., Gardner and Ashmore, 2011). The output of the batch processing script was an orthophoto and a DEM of the flume surface with 1.5 mm pixels. The script additionally exported a report on the PhotoScan project, indicating the number of photos used, the image overlap, and the estimated error on target detection. Morgan et al. (2016) reported that they were unable to utilize the automated target detection feature in PhotoScan but our fixed geometry and consistent survey method may have been important to successful automated target identification. Furthermore, it was found that using large identification values on the targets (e.g., ID numbers > 100 (see Fig. 1)) helped to avoid PhotoScan confusing targets signals during processing, which also improved overall batch processing success.

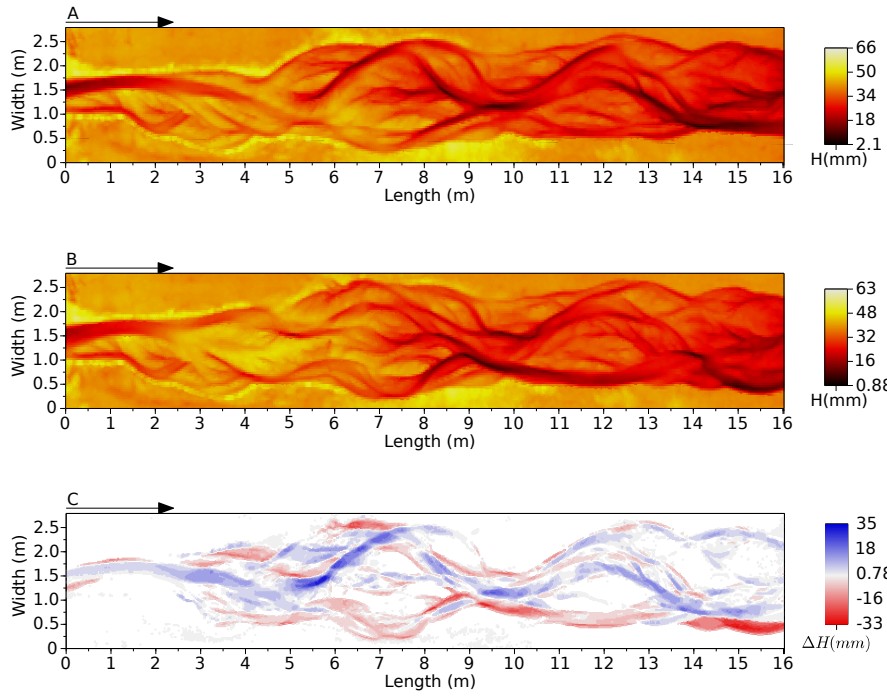

**Figure 5.** Generation of a DEM of Difference (DoD) using two consecutive DEMs where (A) DEM2 (time = 1200 min) was subtracted from (B) DEM1 (time = 1400 min) to create a (C) DoD where areas of erosion are red and areas of deposition are blue.

## 5  Applications in Braided River Geomorphology

### 5.1  DEMs and DEMs of Difference

Examples of the final DEMs and DEMs of difference (DoDs) are shown in Fig. 5. From the batch processed DEMs there was flexibility in post-processing (using custom Scilab scripts) for cropping, filtering, and change detection thresholds for various geomorphic analyses including extracting both the areas and volumes of erosion and deposition. An alternative to using custom scripts would be the software program Geomorphic Change Detection (Wheaton et al., 2010a, b) or ArcGIS, although a comparison of techniques was not completed for this research. In addition to estimating changes in the morphological active width (Peirce et al., 2018a), the data can be used for many applications, including estimates of water depth and bed surface texture.

### 5.2  Water Depth

Estimation of water depths and water surface slope is valuable in small scale models of complex planform where direct measurement and synoptic mapping of water depth are very difficult due to shallow ($\sim$ 2 cm) depths and constantly changing

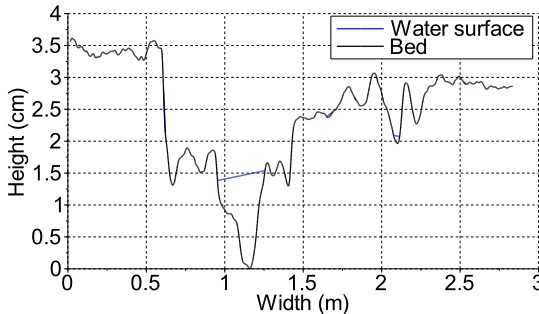

**Figure 6.** Example of water surface detection. In the cross section, we assume a straight line between the first water point (i.e. the difference is higher than the threshold) on the cross section and the first next dry point (i.e. the difference is lower than the threshold).

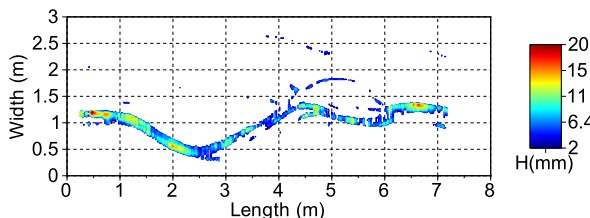

**Figure 7.** Example of a water depth map for a small (8m) reach of the flume.

morphology. Detection of the water surface was possible by creating a DoD from the water surface DEM and matching dry bed DEM for a given time interval (Fig. 6).

The photo surveys of the water surface were taken in the final minute of each 15 minute run to avoid morphological change between the subsequent dry bed photo surveys. In the DoD, only change greater than 1 mm (approximate mean error through all the experiments, Table 2) was considered and it was assumed that differences detected were the result of water depth and not morphological change (Fig. 6). From this data, a binary map (water/non water) was created and the elevation of the water on a cross-section was taken to be the elevation of the closest non-water cell. For this analysis, it was assumed that the water level was straight at the cross-sectional scale, and a water depth map was extracted from the cross section analysis (Fig. 7).

The technique used requires refinement and further assessment but presents an important area for future development of SfM methods in laboratory models and flumes.

Validation is currently problematic because water depth is not measured directly. This could be improved if it was possible to make a direct comparison between water surface and bed elevation at a point.

## 5.3 Bed Surface Texture

Maps and data of the grain size distribution and bed surface texture can be used for a variety of analyses but their immediate value is in showing the wide range, and spatial patterns, of texture (grain size) on the gravel braided river model bed surface. Previous papers based on experiments in the same flume and sediment have shown that maps of bed surface texture, as a measurement of grain size variation, may be produced from the same imagery created for the photogrammetry (Leduc et al., 2015).

For bed material larger than the cell size, SfM may be an alternative method to estimate the bed grain size using the DEM surface roughness, and correlating the standard deviation of elevations with particle diameters Pearson et al. (2017). This is likely to be less successful when grain roughness is less than the precision of the DEMs and pixel sizes are approximately equal to median particle diameter as in this close-range application in a small-scale flume model.

For the medium sand we used, grain size analysis and mapping was based on the image texture method developed and tested by Carbonneau (2005) and Carbonneau et al. (2005) for field mapping of gravel-bed rivers and previously adapted for the sand texture of physical models (Gardner and Ashmore, 2011; Leduc et al., 2015). The image texture calculation was made using the co-occurrence gray matrix level based on 64 gray level vertical bed images. The sampling window size of 7 * 7 pixels was chosen due to the median grain size (1.3 mm) and the camera resolution, and the best fit of the data was found using the entropy index. To calibrate the predictive relationship between an entropy value and the real grain size, two sets of samples were used.

The first was based on the surface grain size samples from the Sunwapta River, Canada (a gravel-bed braided river on which the flume grain size distribution was based) and the second was based on uniform grain size patches from the flume bed material. The field calibration dataset was generated from 13 grain size samples randomly selected from a larger dataset of 30 samples. The field samples were manually sieved using an adaptation of the paint-and-pick technique, where a chalk dot was drawn on every visible surface grain. The 13 grain size samples were downscaled to get the calibration grain size sample composition. In addition to the non-uniform field samples, 6 uniform samples were also created from the different grain sizes of the bed material.

In total, these 19 samples covered the full range of grain size in the flume (Fig. 8 A). For the flume calibration dataset, grains were mixed and glued to a white foam board in a continuous thick layer with an area of 10 cm * 15 cm (Fig. 8 B). The sampling board was placed on the flume bed at different locations and the entropy value was estimated for each sample at each location over a square of $\sim 3.7 cm^2$. Each sample represented over 5000 pixels on the picture. The final flume calibration relationship was built using the median grain size of the sample and the corresponding entropy value (Fig. 9).

In addition to the calibration datasets, a validation dataset was created from 100 grain samples on the flume bed, collected using $1 cm^2$ wax drops poured directly onto the bed (Fig. 10) and manually sieved. On the corresponding orthophotos (Fig.10), the centre of each wax drop was manually set and the entire wax drop surface was automatically detected using a color threshold. Of the initial 100 wax samples, 70 were used for validation. The measured grain size was compared to the entropy maps generated from the texture calculation (Fig. 11) and for which the mean absolute error was 0.02 mm with a standard deviation of 0.48 mm, although the relative error was higher for smaller grains.

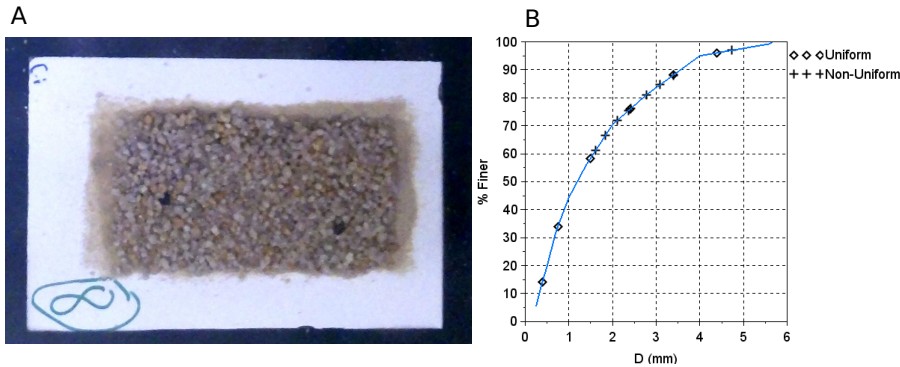

**Figure 8.** (A) An example of calibration sample used during the calibration of the bed surface texture, and (B) the median grain size of the calibration data samples, including uniform and non-uniform samples. The line is the flume grain size distribution.

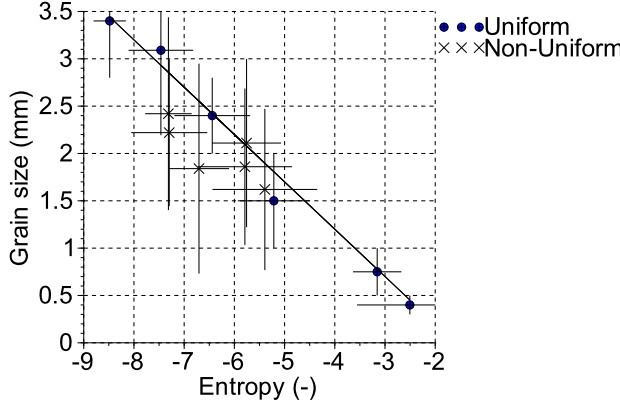

**Figure 9.** The grain size calibration for uniform and non-uniform samples. Horizontal error bars are the entropy standard deviation of the sample and the vertical bars are the grain size sieving range.

We refer to the estimated grain size from the textural calibration as the "equivalent texture" because it is a texture value calibrated to only the median grain size (not the full distribution) for a patch and is not strictly a grain size value as conventionally defined in physical measurements of grain size. A final grain size map was created for each DEM, mapping the bed elevation and local bed texture for the entire model surface (Fig. 12). Combining grain maps derived from orthoimagery, with DEMs and

5  DoDs can provide data on, for example, bed roughness and changes over time, grain size sorting for sedimentological analysis and relations to bed morphology, and relationship to topographic roughness.

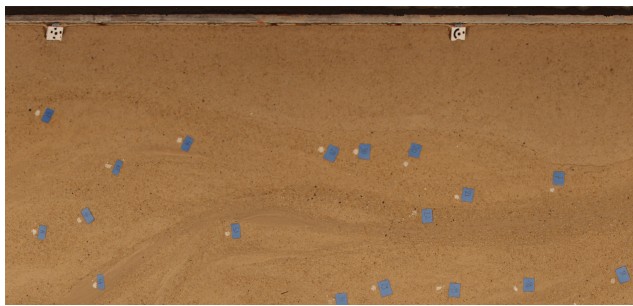

**Figure 10.** Image of the flume surface with the location of wax samples used for grain size validation.

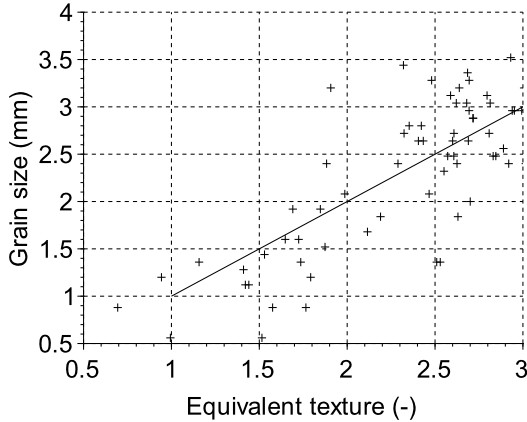

**Figure 11.** The validation dataset: the grain size from the texture analysis as compared to the hand-selected grain size.

## 5.4 Application in geomorphology analysis

The SfM technology applied to our flume experiments provides more than 1000 DEMs and orthophotos which leads to an extensive geomorphological processes analysis. Based on these DEMs, studies focusing on the active width, planform evolution, grain size distribution and variability at a high temporal resolution (see Middleton et al. (2018); Peirce et al. (2018a, b)) are providing new insights on braided river morphology, dynamics and bedload transport.

## 6 Conclusions

This paper presented methods for the successful application of SfM photogrammetry using AgiSoft PhotoScan in a physical model of a gravel-bed braided river. Consideration of camera geometry, automated control-target detection, image quality, and batch processing made it possible to create a large number (>1000) of high-resolution DEMs of complex braided channel morphology with vertical precision on the order of 1 mm. These DEMs can be used extensively, including to map and quantify

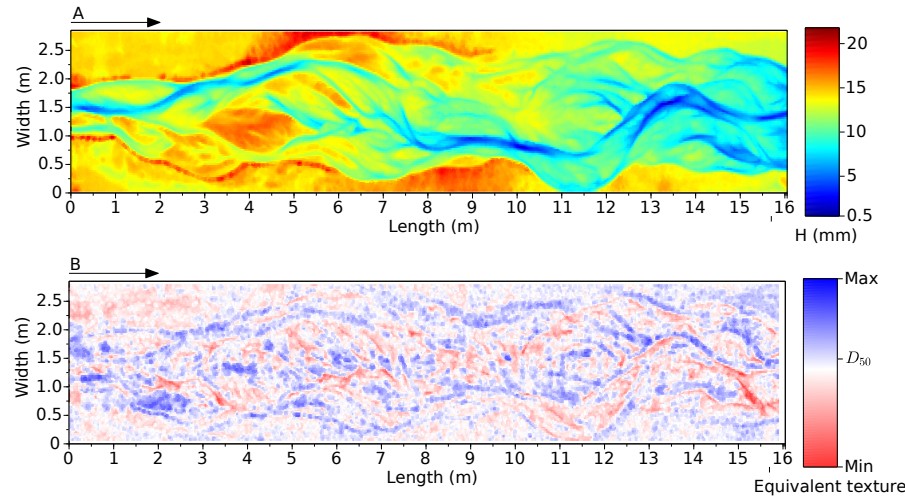

**Figure 12.** An example DEM (A) plotted with its associated equivalent texture map (B).

morphological change (using DEMs of difference) as well as to acquire water surface DEMs to map wetted areas and estimate water depth. Additionally, the images collected can be used for mapping grain size variation across the braided river. The results presented demonstrate that SfM can yield large volumes of very high quality topographic data efficiently in close-range laboratory applications. In this way we have extended collective learning about the quality of SfM data acquisition methods in this type of laboratory setting and model (Morgan et al., 2016) and added to the range of conditions to which this technology has been applied.

*Data availability.*  The data presented in the figures of this paper are available from the corresponding author.

*Acknowledgements.*  This research was supported by a Natural Sciences and Engineering Research Council Discovery Grant (41186-2012) to P. Ashmore. Flume construction was supported by the Canada Foundation for Innovation and Newalta Resources Inc. Additional support was provided by the Vanier Canada Graduate Scholarship awarded to S. Peirce. Thank you to everyone who assisted with data collection, especially L.Middleton and D. Barr.

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
