# Peer review of "Short Communication: Challenges and Applications of Structure-from-Motion Photogrammetry in a Physical Model of a Braided River"

_Earth Surface Dynamics, 2018_

## Referee Comment (RC1) · Anonymous Referee #1 · 27 Jun 2018

The introduced study describes the application of SfM to measure DEMs of flumes in laboratory setups. Images for SfM are acquired in sequence and resulting DEMs are compared to each other and to TLS data. The manuscript is well written and clearly structured. Methods and results are illustrated sufficiently. However, there are some concerns, which should be addressed before the manuscript can be accepted for publication. The novelty of the introduced results seems to be not very high because many of the mentioned findings, e.g. regarding doming effects or the impact of image quality, are already discussed in other work but just for different scales (e.g. see James et al.

[Figure]

2017, Mosbrucker et al. 2017). The generation of a very high number of DEMs used for DoD calculation is interesting and could be novel if the potential of such data regarding the expected new insights into investigating fluvial processes would be displayed and discussed. Furthermore, the processing of such data to extract the relevant information needed to assess the processes would be of interest. Regarding the references, some more literature should be included concerning the utilization of SfM in laboratory setups in geomorphological applications. For instance, Galland et al. 2015 use SfM and time-lapse imagery in geological experiments with sub-mm accuracy, Kaiser et al. 2018 as well achieve sub-mm accuracy when they perform close-range SfM measurements to detect soil surface changes, and Balaguer-Puiga et al. 2017 use SfM to measure soil erosion at micro-plots in the lab. Furthermore, some concerns exist regarding the usage of two sets of images to estimate the error in this study due to the missing consideration of spatial correlation of errors. Please, see a more detailed description of the raised concerns in the specific comments section.

Specific comments: p. 1 l. 14-19: The DEMs are not mainly limited by the image quality. There are further important error sources leading to potential systematic errors (e.g. dome effects) as well as to random errors, which are highly spatially correlated, amongst others due to the right choice of parameters and their setting (see James et al. 2017).

p. 1 l. 21: More literature regarding SfM and fluvial morphology should be introduced, e.g. Javernick et al. 2014 or Woodget et al. 2015. These authors are one of the first to introduce SfM (in combination with UAV) to fluvial morphology.

p. 2 l. 18-20: I am afraid that I do not understand in what sequence the image pairs were acquired. Were the two sets of images taken during one acquisition (thus both images in short sequence at each position) or were two acquisitions performed in sequence (thus images once during first interval and once again during second interval)? This information would be important because if the images were acquired from the same position just in sequence their suitability to asses DEM errors would be ques-
tionable because acquisition geometry would be almost identical and thus not much change expected in the images. Generally, if camera geometry and surface texture conditions (also considering lighting) for both sets of digital images are similar, not much information regarding accuracy, utilizing DoD differencing, can be expected because errors are spatially highly correlated (James et al. 2017). The raised concern regarding spatial error correlation also relates to p. 4 l. 7-9

chapter 3.1: Why did the authors not exclude some of the coded targets (because many are given) during the bundle adjustment so these targets could be considered as check points and thus used for accuracy assessment of each SfM surface and camera geometry reconstruction?

p. 4. l. 3: What TLS has been used? What accuracy and resolution does the device provide?

p. 4 l. 9-10: The usage of just one value (mean of entire DoD) is not able to describe the spatially variable error, e.g. due to potential tilting. How is this considered for the decision of the DEM?

p. 4 l. 10-12: How certain are the authors that surface changes to the previous time interval are not conflicting the decision for the most suitable DEM of the subsequent interval?

p. 6 l. 3: Already James et al. 2017b illustrate the importance of GCP number and distribution for the DEM quality. Maybe refer to their work.

p. 6 l. 4-8: Please, refer to James & Robson 2014 regarding doming effect because they perform extensive simulations to explain the causes (i.e. image geometry) of doming errors and already show that convergent images improve data accuracy.

p. 6. l. 9: Please, refer to Mosbrucker et al. 2016 who explain very detailed the importance of image quality for DEMs derived with SfM.

p. 6. l. 11: Why is the fixed focal length essential during low light conditions and

low texture? The interior geometry does not influence these circumstances. The fixed focal length is important regarding a reliable camera self-calibration. Good texture is essential for feature extraction and matching but not influenced by the stability of the focal length. To improve texture e.g. aperture and/or exposure time should be adapted (see Mosbrucker et al. 2016 for much more detail).

p. 7 l. 1-2: How was the DEM interpolated from the dense point cloud? PhotoScan offers different options potentially influencing the final DEM.

chapter 5.2: How certain are the authors that indeed water surface has been detected/reconstructed with SfM? The "water surface" could also be the result of some interpolation artefact in PhotoScan because the water is moving and thus feature matching in this area from images captured in (although very short) temporal sequence is unreliable. Did the authors perform some independent reference measurement of the water depth to confirm the SfM results?

p. 8 l. -10: I am afraid that I do not understand what is meant by cross-sectional scale? Did the authors extract water levels at each cross section? If yes, how were the cross-sections extracted and what would be the spatial resolution?

chapter 5.3: Maybe, the authors could also test the usage of the retrieved 3D data with SfM to extract grain sizes directly from roughness estimates calculated with the DEMs. Kaiser et al. 2015 and Pearson et al. 2017 illustrate the great potential of SfM for this task. Furthermore, the authors might also refer to Woodget et al. 2018 regarding the usage of image texture and grain size estimation concerning most recent efforts in this regard because they use the original image instead of the potentially interpolated (and thus introducing further uncertainty) orthophoto.

Figures: The figures involving flume display are very small and thus difficult to read and interpret.

References:

Balaguer-Puiga, Matilde, Ángel, Marqués-Mateua, José Luis Lermaa, Sara, Ibáñez-Asensio (2017): Estimation of small-scale soil erosion in laboratory experiments with Structure from Motion photogrammetry, Geomorphology, 295

Galland Olivier, Håvard S. Bertelsen, Frank Guldstrand, Luc Girod, Rikke F. Johannessen, Fanny Bjugger, Stef fi Burchardt, and Karen Mair (2016): Application of open-source photogrammetric software MicMac for monitoring surface deformation in laboratory models, Journal of Geophysical Research: Solid Earth

James, M. R. and Robson, S. (2014): Mitigating systematic error in topographic models derived from UAV and ground-based image networks, ESPL, 39

James, M., Robson, S., Smith, M. (2017): 3-D uncertainty-based topographic change detection with structure-from-motion photogrammetry: precision maps for ground control and directly georeferenced surveys, ESPL, 42(12)

James, MR, Robson, S, d'Oleire-Oltmanns, S & Niethammer, U (2017): Optimising UAV topographic surveys processed with structure-from-motion: ground control quality, quantity and bundle adjustment, Geomorphology, 280

Javernick, L., Brasington, J., and Caruso, B. (2014): Modeling the topography of shallow braided rivers using Structure-from-Motion photogrammetry, Geomorphology, 213

Kaiser, A., Neugirg, F., Haas, F., Schmidt, J., Becht, M., and Schindewolf, M. (2015): Determination of hydrological roughness by means of close range remote sensing, SOIL, 1

Kaiser, Andreas, Annelie Erhardt, Anette Eltner (2018): Addressing uncertainties in interpreting soil surface changes by multitemporal high ‐ resolution topography data across scales, LDD

Mosbrucker, Adam R., Jon J. Major, Kurt R. Spicer, John Pitlick (2017): Camera system considerations for geomorphic applications of SfM photogrammetry, ESPL, 42
Pearson E., M.W. Smith, M.J. Klaar, L.E. Brown (2017): Can high resolution 3D topographic surveys provide reliable grain size estimates in gravel bed rivers? Geomorphology, 293

Woodget, A. S., Carbonneau, P. E., Visser, F., and Maddock, I. P. (2015): Quantifying submerged fluvial topography using hyperspatial resolution UAS imagery and structure from motion photogrammetry, ESPL, 40

Woodget, A., Fyffe, C., Carbonneau, P. (2018): From manned to unmanned aircraft: Adapting airborne particle size mapping methodologies to the characteristics of sUAS and SfM, ESPL, 43
* * *

---

## Referee Comment (RC2) · Anonymous Referee #2 · 3 Aug 2018

In this short communication, the authors detail Structure-from-Motion photogrammetry methods related to topographic measurements in a braided river flume experiment. The authors utilize automated batch processing to expedite creation of digital elevation models (DEMs) and provide a sampling of potential further analyses including the calculation of erosion and deposition using DEMs of difference (DoDs) and estimation of water depths. This study extends previous research on using Structure-from-Motion photogrammetry in laboratory flume settings and provides important insight that is relevant for researchers involved in similar physical experiments. The paper is straightfor-

ward, logically organized, and easy to read. However, there are a few issues that need clarification or addressing.

My primary concern is with the "error quantification" in Section 3.1. In subsection 3.1.1 DEMs derived from duplicate photosets of the same surface are compared to "estimate the mean and standard deviation of the vertical error" (P4, L7), while the comparisons of non-changing areas in subsection 3.1.2 are used to "estimate vertical precision" (P4, L15). I would consider the former to be a measure of precision also, rather than "error." The use of the term "error" conveys the idea of comparison to a standard, or a measure of "trueness", while these comparisons are between two surfaces of unknown accuracy. Subsection 3.1.3 does provide potential for actual error estimation, but the reported accuracy of the hand-held laser is not stated. A rewording of the parameters being estimated and quantified by the authors could strengthen section 3.1. I have more comments related to this section that will be included below.

Other comments:

P2, L13: Please also include the geometric standard deviation of the grain size distribution.

P3, L6: The guidance I have seen suggests having stationary lighting sources rather than one that moves with the camera (e.g., the camera flash). This does not seem to have negatively affected your results, but it is counter to general guidelines.

P3, L18: Was there general consistency in the density of the SfM point clouds? How did the point spacing compare to the DEM cell size and what was the interpolation method used to generate the DEMs?

P4, L6: Please clarify, were the two photosets each made up of $\sim$ 100 photos (mentioned in P2, L24)?

P4, L9: Was there a spatial pattern to the differences in the DoD maps (e.g., greater differences in areas with more complex topography)?

P4, L12: Were there any steps taken to ensure that the comparison to the DEM from the previous time did not include an area where geomorphic change may have taken place?

P4, L18: The analysis in section 3.1.1 seems to be a better estimation of the "overall DEM noise" as the entire DEMs were used (< 1 mm, Table 1). Section 3.1.2 is a more localized analysis of DEM noise, where the greater variability (~1 mm, Table 2) may be attributable to the featureless nature of the areas in the images used to generate the elevations of those "non moving, flat areas". The analysis is this section does nicely highlight the effect of data collection improvement by the reduction in mean differences in Table 2.

P4, L21: What is the manufacturer/model of the laser scanner? What is its reported accuracy?

P4, L22: How were the scanner data oriented in real world coordinates? How did the point density from the laser scanner compare with SfM point density?

P5, L1: Was there any spatial pattern to the differences in the DEMs? What was the nature of the 30 cm x 39 cm area scanned (e.g., with or without channels/complex topography)?

P6, L5: What were Photoscan's estimates for target errors? Were they consistent through time, or did they also improve?

P6, L14: The combining of DEMs described in subsection 3.1.1 is not derived from a single set of images. I'm not sure the last sentence of this paragraph is necessary or meaningful for how the data were processed.

P6, L17: What are the specs of the machine used for processing (e.g., CPU, RAM)?

P7, L13: I suggest citing Wheaton et al. (2010a) and/or Wheaton et al. (2010b) in reference to Geomorphic Change Detection.

P8, L6: Here you say images were collected in final minute of each experiment, but earlier (P6, L16) you say it took 15 minutes to collect the imagery?

P8, L10: How did derived depth maps compare with visual observations? Figure 7 looks like a single-thread channel. Was that the condition of the flume, or were there many other threads below the threshold of detection?

P8, L11: Possibly make a recommendation or two for future development to improve your method.

P9, Figure 8B: Consider presenting the grain size data as a semi-log plot.

P12, L3: Please consider making your processing scripts (Python and Scilab) available also. You may be interested in also creating an entry on your methods/setup/equipment on Sediment Experimentalist Network (SEN) Knowledge Base (http://sedexp.net/).

Editorial comments:

P1, L22: "recent reports show the SfM techniques…" should read "recent reports show that SfM techniques…"

P2, L12: "2.71 s-1" should be "2.71 m3s-1"

P4, Table 1 caption: "duplicates DEM" should be "duplicate DEMs"

P5, Table 2 caption: I think "Vertical precision" would be a more accurate description than "vertical error"

P5, L7: "Table 1" should be "Table 2"

P6, L12: "the focus as improved" should be "the focus was improved"

P9, L17: "different grain size" should be "different grain sizes"

P11, L5: "precision of the order" should be "precision on the order"

References:

Wheaton, J. M., J. Brasington, S. E. Darby, and D. A. Sear (2010a), Accounting for uncertainty in DEMs from repeat topographic surveys: improved sediment budgets, Earth Surface Processes and Landforms, 35 (2), 136-156, doi:10.1002/esp.1886.

Wheaton, J. M., J. Brasington, S. E. Darby, J. Merz, G. B. Pasternack, D. Sear, and D. Vericat (2010b), Linking geomorphic changes to salmonid habitat at a scale relevant to fish, River Research and Applications, 26 (4), 469-486, doi:10.1002/rra.1305.

---

## Author Comment (AC1) · 12 Oct 2018

We thank the two referees for their comments. Our responses below are organised to respond to each review in sequence.

We are planning to make the following changes :

- New references will be added
- Technical details will be added (calculations, interpolation, laser scan,...)
- The error/precision section will be clarified

*Answers are in italic font.*

**Anonymous Referee 1**

The introduced study describes the application of SfM to measure DEMs of flumes in laboratory setups. Images for SfM are acquired in sequence and resulting DEMs are compared to each other and to TLS data. The manuscript is well written and clearly structured. Methods and results are illustrated sufficiently. However, there are some concerns, which should be addressed before the manuscript can be accepted for publication. The novelty of the introduced results seems to be not very high because many of the mentioned findings, e.g. regarding doming effects or the impact of image quality, are already discussed in other work but just for different scales (e.g. see James et al. 2017, Mosbrucker et al. 2017).The generation of a very high number of DEMs used for DoD calculation is interesting and could be novel if the potential of such data regarding the expected new insights into investigating fluvial processes would be displayed and discussed. Furthermore, the processing of such data to extract the relevant information needed to assess the processes would be of interest. Regarding the references, some more literature should be included concerning the utilization of SfM in laboratory setups in geomorphological applications. For instance, Galland et al. 2015 use SfM and time-lapse imagery in geological experiments with sub-mm accuracy, Kaiser et al. 2018 as well achieve sub-mm accuracy when they perform close-range SfM measurements to detect soil surface changes, and Balaguer-Puiga et al. 2017 use SfM to measure soil erosion at micro-plots in the lab. Furthermore, some concerns exist regarding the usage of two sets of images to estimate the error in this study due to the missing consideration of spatial correlation of errors. Please, see a more detailed description of the raised concerns in the specific comments section.

*Thank you for your comments. Indeed, the SfM is providing a large amount of data but the main focus of the paper is SfM application to laboratory flumes rather than morphology studies. Three papers have been recently published based on the data-set [Peirce et al., 2018a,b, Middleton et al., 2018] which provide examples of applications to analysis of relevant fluvial processes. We can add to the paper example of results and geomorphic analysis. The comments regarding references will be addressed on the specific comments.*

*S. Peirce, P. Ashmore, and P. Leduc. The variability in the morphological active width: Results from physical models of gravel-bed braided rivers.* Earth Surface Processes and Landforms, *43(11):2371–2383, may 2018a. doi: 10.1002/esp.4400.*

*S. Peirce, P. Ashmore, and P. Leduc. Evolution of grain size distributions and bed mobility during hydrographs in gravel-bed braided rivers.* Earth Surface Processes and Landforms, *sep 2018b. doi: 10.1002/esp.4511.*

*L. Middleton, P. Ashmore, P. Leduc, and D. Sjogren. Rates of planimetric change in a proglacial gravel-bed braided river: field measurement and physical modeling.* Earth Surface Processes and Landforms, *oct 2018. doi: 10.1002/esp.4528.*

**Specific comments:**

p. 1 l. 14-19: The DEMs are not mainly limited by the image quality. There are further important error sources leading to potential systematic errors (e.g. dome effects) as well as to random errors, which are highly spatially correlated, amongst others due to the right choice of parameters and their setting (see James et al. 2017).

*We noticed a doming effect during our primary tests but the camera calibration and parameter adjustments reduced it and we didn't notice any obvious systematic error afterwards. Further details will be added to the paper based on the mentioned references.*

p. 1 l. 21: More literature regarding SfM and fluvial morphology should be introduced, e.g. Javernick et al. 2014 or Woodget et al. 2015. These authors are one of the first to introduce SfM (in combination with UAV) to fluvial morphology.

*Further detail will be added to the paper based on the mentioned references. Note that we limited the review to those applications specifically related to close-range applications in laboratory flumes with fixed geometry which is the focus of the paper, rather than more broadly to fluvial morphology primarily acquired from drones in the field which introduce other analytical issues less relevant to our work.*

p. 2 l. 18-20: I am afraid that I do not understand in what sequence the image pairs were acquired. Were the two sets of images taken during one acquisition (thus both images in short sequence at each position) or were two acquisitions performed in sequence (thus images once during first interval and once again during second interval)? This information would be important because if the images were acquired from the same position just in sequence their suitability to asses DEM errors would be questionable because acquisition geometry would be almost identical and thus not much change expected in the images. Generally, if camera geometry and surface texture conditions (also considering lighting) for both sets of digital images are similar, not much information regarding accuracy, utilizing DoD differencing, can be expected because errors are spatially highly correlated (James et al. 2017). The raised concern regarding spatial error correlation also relates to p. 4 l. 7-9.

*The 2 sets of pictures were taken on the same surface but during 2 different acquisitions. The second set of pictures was taken after the first one was completely done, which means they are two separate traverses of the camera trolley along the flume length. The 2 sets of pictures don't have the same number of pictures, mean overlap, or exact start and end locations. This gives at least some estimate of the precision and repeatability of the survey. Details can be added to the paper.*

chapter 3.1: Why did the authors not exclude some of the coded targets (because many are given) during the bundle adjustment so these targets could be considered as check points and thus used for accuracy assessment of each SfM surface and camera geometry reconstruction?

*We contemplated this but some targets weren't well detected on every DEM, especially during the early experiments. We preferred to keep the entire target set and try a different way to estimate the error and precision using the model surfaces rather than a few targets.*

p. 4. L. 3: What TLS has been used? What accuracy and resolution does the device provide?

*The scanner is a hand-held Exascan scanner from Creaform for which distance to the surface could be kept relatively constant rather than the radial distance effects of TLS at very close range. The resolution is 0.050 mm and the accuracy up to 0.040 mm. This information will be added to the paper.*

p. 4 L. 9-10: The usage of just one value (mean of entire DoD) is not able to describe the spatially variable error, e.g. due to potential tilting. How is this considered for the decision of the DEM?

*Indeed, the mean value isn't able to describe a potential tilting; nevertheless we didn't noticed any consistent spatial variability (see figure below) on the DoD or a tilting on the cross section or longitudinal profiles.*

p. 4 l. 10-12: How certain are the authors that surface changes to the previous time interval are not conflicting the decision for the most suitable DEM of the subsequent interval?

*The two DEMs for each time interval are generated by the same process each time. They are therefore detecting the same changes from the previous DEM so that both DoDs contain the (same) real morphological change as well as the DEM error. Our method was intended to include potential differences due to DEM error and to select the DEM for which the 'global' errors were smallest. We are assuming that the DEM error will add topographic bed variation and so increase the mean value.*

p. 6 l. 3: Already James et al. 2017b illustrate the importance of GCP number and distribution for the DEM quality. Maybe refer to their work.

*We will refer to their work.*

p. 6 l. 4-8: Please, refer to James and Robson 2014 regarding doming effect because they perform extensive simulations to explain the causes (i.e. image geometry) of doming errors and already show that convergent images improve data accuracy.

*We will refer to their work.*

p. 6. l. 9: Please, refer to Mosbrucker et al. 2016 who explain very detailed the importance of image quality for DEMs derived with SfM.

*We will refer to their work.*

p. 6. l. 11: Why is the fixed focal length essential during low light conditions and low texture? The interior geometry does not influence these circumstances. The fixed focal length is important regarding a reliable camera self-calibration. Good texture is essential for feature extraction and matching but not influenced by the stability of the focal length. To improve texture e.g. aperture and/or exposure time should be adapted (see Mosbrucker et al. 2016 for much more detail).

*We can rephrase this to reflect the point. The fixed focal length is useful at close range (not relevant for UAV imagery) to keep the focus as sharp and consistent as possible which has a major effect on the quality of the results if low light affects the auto-focus.*

p. 7 l. 1-2: How was the DEM interpolated from the dense point cloud? PhotoScan offers different options potentially influencing the final DEM.

*We use the Photoscan interpolation (enabled option).*

chapter 5.2: How certain are the authors that indeed water surface has been detected/ reconstructed with SfM? The "water surface" could also be the result of some interpolation artefact in PhotoScan because the water is moving and thus feature matching in this area from images captured in (although very short) temporal sequence is unreliable. Did the authors perform some independent reference measurement of the water depth to confirm the SfM results?

*The water surface is not directly detected with SfM. We are actually assuming that water surface disturbs the elevation signal. We compared the DEM created with the water flow, with that from the dry bed DEM and identified cells with elevation differences above a threshold which we are assuming represent the water extent. We are not using the real water running DEM but only the binary maps water/dry resulting from the DoD between the dry and wet DEM. To estimate the water depth, we use the dry topography given the positon of the water edge and extent. Measuring the real water depth is quite difficult except for a few local 'spot checks' but we are able to compare the water surface extent with that visually apparent on the images. This is effective in the case of wide, shallow braided channels but may be less effective for other river morphology. More details will be added to the paper.*

p. 8 l. -10: I am afraid that I do not understand what is meant by cross-sectional scale? Did the authors extract water levels at each cross section? If yes, how were the cross-sections extracted and what would be the spatial resolution?

*The point is that we assume that the water surface is straight for any given cross-section of the river between points where the water surface intersects the bed topography (Fig 6). The water surface estimation*

*is done for each cross-section so that the spatial resolution is the same as that of the DEM.*

chapter 5.3: Maybe, the authors could also test the usage of the retrieved 3D data with SfM to extract grain sizes directly from roughness estimates calculated with the DEMs. Kaiser et al. 2015 and Pearson et al. 2017 illustrate the great potential of SfM for this task. Furthermore, the authors might also refer to Woodget et al. 2018 regarding the usage of image texture and grain size estimation concerning most recent efforts in this regard because they use the original image instead of the potentially interpolated (and thus introducing further uncertainty) orthophoto.

*References will be added to the paper to mention this option but we are not in a position to test the idea. Note that pixel size is of the same order as the D50 of the particle sizes so the textural effects on the DEM may be difficult to detect given the precision of elevation measurement. We also draw attention again to the differences between our small-scale rivers and the full scale UAV-based procedures referred to in the suggested papers.*

**Figures:**

The figures involving flume display are very small and thus difficult to read and interpret.
*These can be enlarged.*

**References:**

Balaguer-Puiga, Matilde, Ángel, Marqués-Mateua, José Luis Lermaa, Sara, Ibáñez- Asensio (2017):, Geomorphology, 295

Galland Olivier, Havard S. Bertelsen, Frank Guldstrand, Luc Girod, Rikke F. Johannessen, Fanny Bjugger, Stef Burchardt, and Karen Mair (2016): Application of open-source photogrammetric software MicMac for monitoring surface deformation in laboratory models, Journal of Geophysical Research: Solid Earth

James, M. R. and Robson, S. (2014): Mitigating systematic error in topographic models derived from UAV and ground-based image networks, ESPL, 39

James, M., Robson, S., Smith, M. (2017): 3-D uncertainty-based topographic change detection with structure-from-motion photogrammetry: precision maps for ground control and directly georeferenced surveys, ESPL, 42(12)

James, MR, Robson, S, d'Oleire-Oltmanns, S and Niethammer, U (2017): Optimising UAV topographic surveys processed with structure-from-motion: ground control quality, quantity and bundle adjustment, Geomorphology, 280

Javernick, L., Brasington, J., and Caruso, B. (2014): Modeling the topography of shallow braided rivers using Structure-from-Motion photogrammetry, Geomorphology, 213

Kaiser, A., Neugirg, F., Haas, F., Schmidt, J., Becht, M., and Schindewolf, M. (2015): Determination of hydrological roughness by means of close range remote sensing, SOIL, 1

Kaiser, Andreas, Annelie Erhardt, Anette Eltner (2018): Addressing uncertainties in interpreting soil surface changes by multitemporal high resolution topography data across scales, LDD Mosbrucker,

Adam R., Jon J. Major, Kurt R. Spicer, John Pitlick (2017): Camera system considerations for geomorphic applications of SfM photogrammetry, ESPL, 42

Pearson E., M.W. Smith, M.J. Klaar, L.E. Brown (2017): Can high resolution 3D topographic surveys provide reliable grain size estimates in gravel bed rivers? Geomorphology, 293

Woodget, A. S., Carbonneau, P. E., Visser, F., and Maddock, I. P. (2015): Quantifying submerged fluvial topography using hyperspatial resolution UAS imagery and structure from motion photogrammetry, ESPL, 40

Woodget, A., Fyffe, C., Carbonneau, P. (2018): From manned to unmanned aircraft: Adapting airborne particle size mapping methodologies to the characteristics of sUAS and SfM, ESPL, 43

**Anonymous Referee 2**

In this short communication, the authors detail Structure-from-Motion photogrammetry methods related to topographic measurements in a braided river flume experiment. The authors utilize automated batch processing to expedite creation of digital elevation models (DEMs) and provide a sampling of potential further analyses including the calculation of erosion and deposition using DEMs of difference (DoDs) and estimation of water depths. This study extends previous research on using Structure-from-Motion photogrammetry in laboratory flume settings and provides important insight that is relevant for researchers involved in similar physical experiments. The paper is straightfor-ward, logically organized, and easy to read. However, there are a few issues that need clarification or addressing. My primary concern is with the "error quantification" in Section 3.1. In subsection 3.1.1 DEMs derived from duplicate photosets of the same surface are compared to "estimate the mean and standard deviation of the vertical error" (P4, L7), while the comparisons of non-changing areas in subsection 3.1.2 are used to "estimate vertical precision" (P4, L15). I would consider the former to be a measure of precision also, rather than "error." The use of the term "error" conveys the idea of comparison to a standard, or a measure of "trueness", while these comparisons are between two surfaces of unknown accuracy. Subsection 3.1.3 does provide potential for actual error estimation, but the reported accuracy of the hand-held laser is not stated. A rewording of the parameters being estimated and quantified by the authors could strengthen section 3.1. I have more comments related to this section that will be included below.

*Thank you for your comments. We will clarify the error/precision section.*

**Other comments:**

P2, L13: Please also include the geometric standard deviation of the grain size distribution.
*The geometric standard deviation of the grain size distribution is 1,4 mm. It will be added to the paper.*

P3, L6: The guidance I have seen suggests having stationary lighting sources rather than one that moves with the camera (e.g., the camera flash). This does not seem to have negatively affected your results, but it is counter to general guidelines.
*At first we considered using a stationary light as you mentioned but the flume is very close (less than a meter) to a white wall reflecting light. The resulting light would be not constant over the flume width and it is very difficult to get uniform light over an interior surface that is 3 x 18m. Previous experience demonstrate the shortcomings of this approach and we had more success with lights that move with the cameras to get a uniform and consistent light.*

P3, L18: Was there general consistency in the density of the SfM point clouds? How did the point spacing compare to the DEM cell size and what was the interpolation method used to generate the DEMs?
*The order of magnitude of the point cloud density was 80 points/$cm^2$,which correspond to 80 points for 45 DEM cells. We didn't notice density variations over time or space. We use the Photoscan interpolation (enabled option).*

P4, L6: Please clarify, were the two photosets each made up of 100 photos (mentioned in P2, L24)?
*Yes, each photo-set is made of 100 photos. We will clarify in the text.*

P4, L9: Was there a spatial pattern to the differences in the DoD maps (e.g., greater differences in areas with more complex topography)?
*We didn't notice any consistent spatial pattern on the DoD. The DoD standard deviation seems to be linked to the picture quality (including picture overlap) rather than the bed complexity. Fig 1 (below) shows the mean difference on the DoD maps regarding to the bed standard deviation (we roughly consider that a smooth bed is likely associated to a low standard deviation and a complex bed is associated to a high standard deviation). The random shape of the point cloud indicates that there is no obvious trend between the DoD and the bed complexity. Furthermore, Figures 2 and 3 show an example of 2 different types of bed and the DoD associated, a smooth bed (Fig 2) and a complex bed (Fig 3). The Dod amplitude is in the order of magnitude of $d_{50} = 1.3mm$ (Fig. 2c and 3c). Dods (Fig. 2b and 3b) don't show any spatial pattern but subdued strips*

*(Fig. 3c) may be related to image overlap along the flume.*

P4, L12: Were there any steps taken to ensure that the comparison to the DEM from the previous time did not include an area where geomorphic change may have taken place?

*To choose a DEM we are using the DoD with the previous DEM. The DoD includes both the geomorphic changes and the measurement error – it is the nature of the experiment. The main hypothesis is that the error measurement would widen the elevation values of the DoD rather than narrow it. The chosen DEM is visually checked on the DoD and Dem plots.*

P4, L18: The analysis in section 3.1.1 seems to be a better estimation of the "overall DEM noise" as the entire DEMs were used ($< 1$ mm, Table 1). Section 3.1.2 is a more localized analysis of DEM noise, where the greater variability (1 mm, Table 2) may be attributable to the featureless nature of the areas in the images used to generate the elevations of those "non moving, flat areas". The analysis is this section does nicely highlight the effect of data collection improvement by the reduction in mean differences in Table 2.

*We can revise the text to point this out.*

P4, L21: What is the manufacturer/model of the laser scanner? What is its reported accuracy?

*The TLS used is a Exascan scanner from Creaform. The resolution is 0.050 mm and the accuracy up to 0.040 mm. Those data will be added to the paper.*

P4, L22: How were the scanner data oriented in real world coordinates? How did the point density from the laser scanner compare with SfM point density?

*The hand-held laser scan point density was about half that of the SfM point cloud. Orientation is relative to the walls of the flume.*

P5, L1: Was there any spatial pattern to the differences in the DEMs? What was the nature of the 30 cm x 39 cm area scanned (e.g., with or without channels/complex topography)?

*The bed 30 cm * 39 cm surface was in the side on the main channel and typical of the model topography in general. There were small bed elevation changes and it included channel margins and banks.*

P6, L5: What were Photoscan's estimates for target errors? Were they consistent through time, or did they also improve?

*The average target error from Photoscan was from 0,005 m to 0,001 m. The error is consistent within each experiment as targets are removed from the side of the flume each time the bed is flatten, ie at the end of each experiment.*

P6, L14: The combining of DEMs described in subsection 3.1.1 is not derived from a single set of images. I'm not sure the last sentence of this paragraph is necessary or meaningful for how the data were processed.

*We can revise accordingly.*

P6, L17: What are the specs of the machine used for processing (e.g., CPU, RAM)?

*The specs of the machine are: 32GB RAM, Intel Core i7 processors (4790k) @ 4 GHz.*

P7, L13: I suggest citing Wheaton et al. (2010a) and/or Wheaton et al. (2010b) in reference to Geomorphic Change Detection.

*We will refer to their work.*

P8, L6: Here you say images were collected in final minute of each experiment, but earlier (P6, L16) you say it took 15 minutes to collect the imagery?

*To estimate the water depth, the 'wet' set of picture was taken few minutes before the end of the 15 minute run. Only a short part of the flume was considered and only a single set of picture was taken. The number of pictures and thus the length on the wetted DEM was related to the time it took to collect the imagery.*

[Figure]

Figure 1: DoD precision versus topographic complexity of the bed

[Figure]

Figure 2: A relatively smooth and simple topography (a) with associated DoD of duplicate DEMs (b) and the Dod distribution (c)

[Figure]

Figure 3: Example of a complex bed (a), the duplicate Dod (b) and the Dod distribution (c)

P8, L10: How did derived depth maps compare with visual observations? Figure 7 looks like a single-thread channel. Was that the condition of the flume, or were there many other threads below the threshold of detection?

*The method used only detect the deepest channels, mainly the active channels but the shallow channels aren't well detected because flow is extremely shallow (a few mm).*

P8, L11: Possibly make a recommendation or two for future development to improve your method.
*We will add recommendations on the grain size mapping and the water surface detection.*

P9, Figure 8B: Consider presenting the grain size data as a semi-log plot.
*Easily done if required but range of particle sizes makes it unnecessary.*

P12, L3: Please consider making your processing scripts (Python and Scilab) available also. You may be interested in also creating an entry on your methods/setup/equipment on Sediment Experimentalist Network (SEN) Knowledge Base (http://sedexp.net/).
*The scripts and data will be available upon request. We will consider the Sediment Experimentalist Network.*

**Editorial comments:**

*All the editorial comments will be done.*

P1, L22: "recent reports show the SfM techniques" should read "recent reports show that SfM techniques"
P2, L12: "2.71 s-1" should be "2.71 m3s-1"
P4, Table 1 caption: "duplicates DEM" should be "duplicate DEMs"
P5, Table 2 caption: I think "Vertical precision" would be a more accurate description than "vertical error"
P5, L7: "Table 1" should be "Table 2"
P6, L12: "the focus as improved" should be "the focus was improved"
P9, L17: "different grain size" should be "different grain sizes"
P11, L5: "precision of the order" should be "precision on the order"

**References:**

Wheaton, J. M., J. Brasington, S. E. Darby, and D. A. Sear (2010a), Accounting for uncertainty in DEMs from repeat topographic surveys: improved sediment budgets,Earth Surface Processes and Landforms, 35 (2), 136-156, doi:10.1002/esp.1886.

Wheaton, J. M., J. Brasington, S. E. Darby, J. Merz, G. B. Pasternack, D. Sear, and D. Vericat (2010b), Linking geomorphic changes to salmonid habitat at a scale relevant to fish, River Research and Applications, 26 (4), 469-486, doi:10.1002/rra.1305.

---

## Referee Report (RR1)

Question: chapter 3.1: Why did the authors not exclude some of the coded targets (because many are given) during the bundle adjustment so these targets could be considered as check points and thus used for accuracy assessment of each SfM surface and camera geometry reconstruction?

*Answer: We contemplated this but some targets weren't well detected on every DEM, especially during the early experiments. We preferred to keep the entire target set and try a different way to estimate the error and precision using the model surfaces rather than a few targets.*

Nevertheless, additional check points improve the error assessment of the SfM model significantly if more targets than necessary are captured and used as check points. Maybe it is worth noticing it in the manuscript.

Question: p. 4 L. 9-10: The usage of just one value (mean of entire DoD) is not able to describe the spatially variable error, e.g. due to potential tilting. How is this considered for the decision of the DEM?

*Answer: Indeed, the mean value isn't able to describe a potential tilting; nevertheless we didn't noticed any consistent spatial variability (see figure below) on the DoD or a tilting on the cross section or longitudinal profiles.*

Maybe the authors should at least consider the standard deviation, as well, to consider systematic effects to some degree?

Question: p. 4 l. 10-12: How certain are the authors that surface changes to the previous time interval are not conflicting the decision for the most suitable DEM of the subsequent interval?

*Answer: The two DEMs for each time interval are generated by the same process each time. They are therefore detecting the same changes from the previous DEM so that both DoDs contain the (same) real morphological change as well as the DEM error. Our method was intended to include potential differences due to DEM error and to select the DEM for which the 'global' errors were smallest. We are assuming that the DEM error will add topographic bed variation and so increase the mean value.*

But what happens if changes occur and one DEM captures them better than the other and shows larger deviations to the previous DEM although they are more precisely than the other DEM with higher error and also lower deviation to the previous DEM because changes are not as well captured? How are such potential cases mitigated?

Question: p. 6. l. 11: Why is the fixed focal length essential during low light conditions and low texture? The interior geometry does not influence these circumstances. The fixed focal length is important regarding a reliable camera self-calibration. Good texture is essential for feature extraction and matching but not influenced by the stability of the focal length. To improve texture

e.g. aperture and/or exposure time should be adapted (see Mosbrucker et al. 2016 for much more detail).

*Answer: We have rephrased this to reflect the point. The fixed focal length is useful at close range (not relevant for UAV imagery) to keep the focus as sharp and consistent as possible which has a major effect on the quality of the results if low light affects the auto-focus.*

But the focus "stability" (?) has nothing to do with the lens type. In zoom as well as fixed lenses focus can be changed. Furthermore, why did the authors use auto-focus at all? This should be avoided especially if the authors use Agisoft lens for prior or posterior calibration. As soon as the focus changes (which can happen often for auto-focus settings) the camera calibration is no more valid. Also, the authors measure the surface from the same distance and thus could set the focus manually.

Question: p. 7 l. 1-2: How was the DEM interpolated from the dense point cloud? PhotoScan offers different options potentially influencing the final DEM.

*Answer: We use the Photoscan interpolation (enabled option).*

Maybe, the authors should discuss a bit more the potential impacts of interpolation error. For instance, was it necessary to interpolate empty cells and how are more than one point within one cell considered?

Further question:

p. 6 l. 17: How does a fixed lens influence the sharpness of an image? It should be possible with a zoom lens, as well.

---

## Author Response (AR2)

We thank the Associate Editor and referees for their comments. Our responses below are organised to respond to each review in sequence.

The font and color code is:

- Black: Comments
- *Italic black: Previous comments and answers*
- Blue: our responses

**Associate Editor comments**

The authors significantly improved the work since the first round of revision. However, some minor issues still remain, as underlined by one of the reviewers.

In addition to the comments raised, I have some suggestions for the figures

1. I would suggest the authors improve the readability of the legend of figure 5c (currently the legend itself doesn't display the colours well)

The legend has been changed

2. figure 6: why is water table sloped? this seems a bit unrealistic

To estimate the water surface, we are working at the cross section scale. But the channels aren't parallel to the flume sides and aren't orthogonal to the cross section. The water table could locally be sloped across the cross section. We'd rather work as it's explained on the paper than arbitrarily set the water level. However, working on transect orthogonal to channel orientation would improve the water table estimate. Also note that the water surface slope is very small – 1mm in 25 cm – but appears steep because of the vertical exaggeration in the figure.

3. figure 11 I think this should read "grain size from the texture analysis as compared to the hand-selected grain size"

The legend has been changed.

**Reviewer comments**

The authors have done a good job revising the manuscript and thus improved their manuscript significantly. All questions have been responded to. However, some (technical) questions still remain open and should be clarified prior considering the manuscript for publication, especially if the contribution is considered as technical note.

*Question: chapter 3.1: Why did the authors not exclude some of the coded targets (because many are given) during the bundle adjustment so these targets could be considered as check points and thus used for accuracy assessment of each SfM surface and camera geometry reconstruction?*
*Answer: We contemplated this but some targets weren't well detected on every DEM, especially during the early experiments. We preferred to keep the entire target set and try a different way to estimate the error and precision using the model surfaces rather than a few targets.*

Nevertheless, additional check points improve the error assessment of the SfM model significantly if more targets than necessary are captured and used as check points. Maybe it is worth noticing it in the manuscript.

We have now added that point to the text

*Question: p. 4 L. 9-10: The usage of just one value (mean of entire DoD) is not able to describe the spatially variable error, e.g. due to potential tilting. How is this considered for the decision of the DEM?*
*Answer: Indeed, the mean value isn't able to describe a potential tilting; nevertheless we didn't noticed any consistent spatial variability (see figure below) on the DoD or a tilting on the cross section or longitudinal profiles.*

Maybe the authors should at least consider the standard deviation, as well, to consider systematic effects to some degree?

We did look at the standard deviation as well as the mean value. The standard deviation was used to target DEMs with an important issue (e.g. tilting, peaks close to the edge of the flume and other artifacts). But we'd rather use the mean value because for an experiment and for DEMs without large error, the range of the standard deviation of the duplicate was smaller than the range of the mean value of the duplicate. We have added some text mentioning this point.

*Question: p. 4 l. 10-12: How certain are the authors that surface changes to the previous time interval are not conflicting the decision for the most suitable DEM of the subsequent interval?*
*Answer: The two DEMs for each time interval are generated by the same process each time. They are therefore detecting the same changes from the previous DEM so that both DoDs contain the (same) real morphological change as well as the DEM error. Our method was intended to include potential differences due to DEM error and to select the DEM for which the 'global' errors were smallest. We are assuming that the DEM error will add topographic bed variation and so increase the mean value.*

But what happens if changes occur and one DEM captures them better than the other and shows larger deviations to the previous DEM although they are more precisely than the other DEM with higher error and also lower deviation to the previous DEM because changes are not as well captured? How are such potential cases mitigated?

Bear in mind that in many cases we are talking about negligible differences and in general we are talking about detecting large systematic errors in one or either of the DEMs relative to the other. Second it is useful to step back and realise that we took this approach mainly to increase the probability of having at least one good DEM for each time period and so to give a continuous time series of topographic change throughout each experiment. Having 2 DEMs to select from allowed us to have at least one high quality DEM when the second DEM had clear errors. The errors could be detected in the DEM data inspection (see above) and in some cases more extensive systematic errors could easily be seen because every DEM had flat sand bed areas along the side of the flume. Error analysis of a larger sample of repeat DEMS from one surface would allow a more thorough evaluation but we do not have the data with which to do that.

*Question: p. 6. l. 11: Why is the fixed focal length essential during low light conditions and low texture? The interior geometry does not influence these circumstances. The fixed focal length is important regarding a reliable camera self-calibration. Good texture is essential for feature extraction and matching but not influenced by the stability of the focal length. To improve texture e.g. aperture and/or exposure time should be adapted (see Mosbrucker et al. 2016 for much more detail).*
*Answer: We have rephrased this to reflect the point. The fixed focal length is useful at close range (not relevant for UAV imagery) to keep the focus as sharp and consistent as possible which has a major effect on the quality of the results if low light affects the auto-focus.*

But the focus "stability" (?) has nothing to do with the lens type. In zoom as well as fixed lenses focus can be changed. Furthermore, why did the authors use auto-focus at all? This should be avoided especially if the authors use Agisoft lens for prior or posterior calibration. As soon as the focus changes (which can happen often for auto-focus settings) the camera calibration is no more valid. Also, the authors measure the surface from the same distance and thus could set the focus manually.

The fixed focal length lens is a general recommendation for photogrammetry going back many years, to keep internal geometry as stable as possible (Mosbrucker et al mention this point also, and other recent papers on SfM in geomorphology), as the original reviewer comment says. The focus is a separate issue, as the reviewer points out. We used manual focus throughout all experiments. We also taped the focus ring, once we had realised part way through the experiments the importance of this additional constraint (also

mentioned in photogrammetry literature including Mosbrucker et al). This is because we also discovered that, on our cameras, the lens focus tended to drift over time even when set manually. Yes, the same could apply if using a zoom lens even if the zoom is fixed. Our point about low light is that the depth of field for very sharp focus may be very limited at large aperture (f 3.5 in our experiments), although this might be mitigated by the short focal length of the lens. The camera-object distance was not constant (varied by 20-30cm over flying height averaging approx. 2.9m) along the length of the flume because the camera rails were horizontal while the was flume tilted at $1.5-2\%$ slope. The major point of the text here is intended to point out the high sensitivity of the results to the sharpness of focus and the need to check this repeatedly and carefully in this type of application. We lost data from two full experiments for this reason alone, even though superficially the focus appeared adequate. Original text: " A fixed focal length lens was essential, especially in low light, close-range conditions, where the surface can have a uniform appearance in photos (Fig. 1) and fixed focal length aids sharpening of focus as well as improving geometric stability (Mosbrucker et al., 2017)." Revised text: "A fixed focal lens is commonly recommended to maximise internal geometric stability (e.g. Mosbrucker et al, 2017.The cameras were also set to a fixed manual focus (rather than auto focus) but focus still slipped slightly at times and we realised that taping the focus ring was necessary. In the low light conditions in the flume, camera aperture had to be large (images were taken at f3.5 and 1/8 second) and this may have reduced depth of field, considering that camera-object distance changed systematically along the flume (because of flume slope) by up to 30 cm. Even when focus was superficially good we discovered problems on the fairly-uniform sand surface and we found that unless close and careful attention was paid to this issue results could be downgraded considerably and large numbers of DEMs could be lost (Fig 4)." In later experiments...

*Question: p. 7 l. 1-2: How was the DEM interpolated from the dense point cloud? PhotoScan offers different options potentially influencing the final DEM.*
*Answer: We use the Photoscan interpolation (enabled option).*
Maybe, the authors should discuss a bit more the potential impacts of interpolation error. For instance, was it necessary to interpolate empty cells and how are more than one point within one cell considered? We didn't look in detail at how Agisoft manages holes in the point cloud or how multiple points are considered. However, visual inspection of selected point clouds show no obvious problems and average point density is very high: $80/cm^2$ .We also looked at the interpolation effect by running two different interpolations on both an almost flat bed (the first DEM with the initial channel carved in the middle of the bed) and on a DEM with a complex morphology. The DoD made with the two different DEMs from the different interpolations shows that interpolation issue are mainly on the flume walls (Fig 1 below). These areas are, in all cases, cropped in our analysis. In the cropped flume areas used in our analysis, the difference between the two types of interpolation is negligible compared to bed morphology and morphological changes (Fig 2).

Further question:
p. 6 l. 17: How does a fixed lens influence the sharpness of an image? It should be possible with a zoom lens, as well.
See our response above.

[Figure]

*Fig 1: Dod between DEMs with different interpolation options (enabled/disabled)*

[Figure]

*Fig 2: Cross section on the initial bed with 2 interpolation options (disabled/enabled)*

[revised manuscript text omitted]